# Laser Er:YAG-Assisted Debonding May Be a Viable Alternative to the Conventional Method for Monocrystalline Ceramic Brackets

**DOI:** 10.3390/ijerph192114564

**Published:** 2022-11-06

**Authors:** Daliana-Emanuela Mocuta, Mariana I. Miron, Diana Lungeanu, Marius Mateas, Emilia Ogodescu, Carmen D. Todea

**Affiliations:** 1Department of Oral Rehabilitation and Dental Emergencies, Faculty of Dentistry, Victor Babeș University of Medicine and Pharmacy, Eftimie Murgu Square No. 2, 300041 Timișoara, Romania; 2Interdisciplinary Research Center for Dental Medical Research, Lasers and Innovative Technologies, 300041 Timișoara, Romania; 3Center for Modeling Biological Systems and Data Analysis, Victor Babeș University of Medicine and Pharmacy, 300041 Timișoara, Romania; 4Department of Functional Sciences, Faculty of Medicine, Victor Babeș University of Medicine and Pharmacy, Eftimie Murgu Square No. 2, 300041 Timișoara, Romania; 5Mechatronics Department, Polytechnic University of Timișoara, 300006 Timișoara, Romania; 6Pediatric Dentistry Research Center, Department of Pediatric Dentistry, Faculty of Dentistry, Victor Babeș University of Medicine and Pharmacy, Eftimie Murgu Square No. 2, 300041 Timișoara, Romania

**Keywords:** monocrystalline ceramic brackets, laser Er:YAG, debonding, laser Doppler flowmetry

## Abstract

In orthodontic practice, due to the increased interest among patients in smile aesthetics, different types of brackets are now being used, with those most frequently applied being ones made of polycrystalline and monocrystalline ceramic. The aim of this study was to evaluate the laser Er:YAG-assisted debonding technique compared to conventional methods for removing monocrystalline ceramic brackets from human teeth. The study sample included 60 vital teeth (frontals of the upper jaw) from 10 patients who had monocrystalline ceramic brackets and were in the final phase of orthodontic treatment. The debonding procedure was carried out following a split-mouth study design, using either the conventional technique or laser Er:YAG 2940 nm radiation. For each tooth, three variables were evaluated: the patient’s sujective tooth sensitivity associated with the debonding, the time required for debonding, and pulp blood flow microdynamics after the debonding. Three evaluation instruments were used to assess and quantify the treatment effects: (i) the Wong–Baker FACES Pain Rating Scale for pain assessment; (ii) a digital stopwatch/timer to measure the time required to remove the bracket; and (iii) laser Doppler flowmetry (LDF) for recording the pulp blood flow evolution. The statistical analysis of the recorded data showed a statistically significant difference between the two debonding methods regarding the tooth sensitivity during the debonding and the time required for the procedure. The subjective tooth sensitivity was reduced from a mean ± standard deviation of 3.07 ± 1.46 to 0.47 ± 0.86 on the Wong–Baker FACES scale (Wilcoxon signed rank, *p* < 0.001). The necessary time for debonding was reduced by 0.697 ± 0.703 s per tooth (paired *t*-test, *p* < 0.001). There was no difference in the blood microdynamics between the two debonding techniques. According to the results of this study, the laser Er:YAG-assisted debonding technique may be a viable alternative to the conventional method for monocrystalline ceramic brackets.

## 1. Introduction

The use of lasers in medicine and surgery has gained widespread acceptance following the advent of the “Ruby Laser”, the first functional laser, built by Theodore Maiman in 1960 [1]. Subsequently, research was directed towards the introduction of laser technology to the field of dentistry. Currently, lasers have many applications, both as a complementary method and as an alternative method to some conventional procedures, such as caries detection, cavity preparation, endodontic treatment, soft tissue surgery, implant decontamination, photobiomodulation, prosthetics, dental materials, etc. [2,3,4,5,6]. In the field of orthodontics, research has been extensively carried out on the effect of laser radiation, which has allowed the development of new therapeutic approaches for tooth movement, pain control, bone regeneration, orthodontic bracket adhesion, and debonding procedures [7,8,9].

Due to the growing interest of patients in smile aesthetics, the use of fixed appliances in orthodontic treatments is an increasingly common practice, both in childhood and adulthood. This necessarily involves fixing brackets to the surface of the tooth enamel at the beginning of the treatment and removing them at the end of the treatment—procedures in which the laser has proven effective [8,9]. In orthodontic practice, several types of brackets can be used, including metal, polycrystalline, and monocrystalline ceramic. The main risk associated with the use of these brackets is the high shear strength at the time of removal, which can cause cracks or permanent damage to the enamel. This shear strength depends mainly on the type of engraving (classic or laser-assisted), the adhesive strength of the resin, the nature of the enamel surface, and the debonding techniques used [10,11].

Despite being aesthetically superior, monocrystalline ceramic brackets have a higher resistance to adhesion to enamel and less to the removal procedure, increasing the risk of fracture due to the low modulus of elasticity, low flexibility, and greater fragility compared to metal brackets [12].

For the debonding of monocrystalline ceramic brackets, special pliers are used, which allow the development of sufficiently large tensile forces to break the adhesion between the bracket and the enamel surface. Side effects such as enamel damage and bracket fracture have been reported frequently after conventional ceramic bracket debonding, depending on the tooth being acted upon [13]. Additionally, the classic method of debonding is associated with painful sensitivity and a thermal effect on the tooth [14,15]; both effects can influence the vascular–nervous package of the tooth, especially the pulpal blood dynamics.

Consequently, in recent years, alternative techniques have been tested in order to remove orthodontic brackets and minimize secondary effects, including ultrasonic or thermal removal [8,12], the use of special pliers or hot air dryers, and the use of laser radiation [9]. Among them, laser systems stand out because they shorten the time required for debonding and reduce the force required for this procedure, and therefore increase comfort for the patient [8,9]. The type of laser device used, the technique of laser radiation, the parameters used, the type of brackets, and the technical characteristics of the brackets must be carefully considered before the debonding procedure in order to avoid potential unwanted side effects [16].

In current practice, the clinical evaluation of the effects of debonding is by clinical inspection of the tooth enamel surface and evaluation of tooth pain sensitivity relayed by the patient, with both being subjective methods. For the evaluation of pulpal vascular microdynamics after dental trauma, according to specialized studies [17,18,19], laser Doppler flowmetry can be used as a non-invasive and objective method of recording pulpal blood flow, which allows the highlighting of preclinical changes in the microcirculation of the dental pulp, both continuously and in real time.

The objective of this clinical study was to investigate the employment of the Er:YAG laser technique for debonding monocrystalline ceramic brackets in comparison with the conventional debonding technique using orthodontic forceps. The three-fold investigation was comprised of the following aspects: (a) the necessary time for debonding; (b) the patient’s subjective tooth sensitivity associated with debonding; and (c) the pulp blood flow microdynamics associated with debonding.

Thus, the following statistical null hypothesis (H_0_) was formulated: There are no statistically significant differences between the Er:YAG laser technique and the conventional one in terms of the debonding time, tooth sensitivity, and the pulp blood flow microdynamics associated with the debonding, when used for the debonding of the monocrystalline ceramic brackets.

## 2. Materials and Methods

The study was carried out within the Discipline of Oral Rehabilitation and Emergencies in Dentistry at the Faculty of Dentistry of the Victor Babeș University of Medicine and Pharmacy in Timișoara. The design and protocol of this study was approved by the Research Ethics Committee (opinion number 21 of 18 March 2022) from the Victor Babeș University of Medicine and Pharmacy in Timișoara, and each patient signed a model informed consent form.

### 2.1. Study Design and Population

A split-mouth study was conducted using a mid-sagittal plane between the maxillary central incisor teeth [19]. It was a two-way factorial design on the treatment and the patient.

Based on previous research that used laser Doppler flowmetry to assess dental pulp vascular micro-dynamics [18] and the expected size of the measurable effect, we calculated the necessary sample size for the following parameters: two treatment groups; alpha = 0.05; beta = 0.1; effect size for both factors = 0.8 (treatment and within-subject factors); 3 + 3 teeth for each patient. It resulted in a total of 60 teeth (10 patients each contributing 3 + 3 teeth), for a power of 0.9.

As the Wong–Baker Pain Rating Scale was also employed, a simulation of the Wilcoxon test’s power level was conducted: 30 teeth in each treatment group, alpha = 0.05; meanA (sdA) = 2 (2); meanB (sdB) = 4 (2). The resulting statistical power was 0.97.

### 2.2. Selection of Subjects

The study included 10 subjects, aged 20–32 years (7 female and 3 male). All subjects had a fixed orthodontic appliance with monocrystalline ceramic brackets applied at least 18 months to 2 years ago. For each subject, we included in the study the six maxillary frontal teeth, which by their position and size allowed for LDF determination. Additionally, we included only the upper frontals to reduce the variability related to the morphology and physiology of the teeth, thus obtaining a more homogeneous sample. Radiographic examination and electrical stimulation of the pulp at the level of the 60 teeth selected in the study confirmed their vitality, without signs of pulpal inflammation (i.e., normal response to electrical stimulation of the pulp, no radiographic evidence of periapical lesions, and a negative response to percussion test). The inclusion criteria included: patients wearing monocrystalline ceramic orthodontic brackets; the lack of analgesic medication for the last 7 days, or other medications that could influence pulpal vascular dynamics and pain perception during the procedure; non-smokers without general health conditions.

### 2.3. Debonding Systems

Orthodontic forceps were used for conventional debonding (Radiance Plus American Ortodontics, Sheboygan, WI, USA).

The laser equipment used for the bracket debonding was the Er:YAG laser (LIGHTWALKER AT S, M021-5AF/ 1S, Fotona d.o.o., Ljubljana, Slovenia) at a wavelength of 2940 nm. It was used together with R14-type fiber, with a cylindrical sapphire tip, 8 mm long and 1.3 mm in diameter, applied perpendicular to the middle of the bracket slot at a distance of 1 mm from the slot (Figure 1). The parameters used were 600 mJ, 2 Hz, a pulse duration of 800 μs, and a fiber diameter of 1.3 mm, with no water-cooling.

### 2.4. Evaluating the Therapeutic Response

Three evaluation instruments were employed to assess and quantify the treatment effects: (i) the Wong–Baker FACES Pain Rating Scale (Figure 2) for pain assessment was used [20,21,22,23,24]. The scale shows six faces, including a happy face, which represents a pain score of 0 and indicates “no hurt”, whereas a crying face represents a pain score of 10 and indicates “hurts like the worst pain imaginable”. The second face represents a pain score of 2 and indicates “hurts a little bit”. The third face represents a pain score of 4 and indicates “hurts a little more”. The fourth face represents a pain score of 6 and indicates “hurts even more”. The fifth face represents a pain score of 8 and indicates “hurts a whole lot”. Based on the faces and their descriptions, the patient chooses the face that best describes their level of pain; (ii) a digital stopwatch/timer (Cromwell, RTL3143120K, Rutland Sport, Nanjing, Jiangsu, China) was used to measure the time required to remove the bracket; and (iii) laser Doppler flowmetry (LDF) for recording the blood flow microdynamics at the dental pulp level [25,26].

To assess the therapeutic response with LDF, the equipment comprised the following: a MoorLab laser Doppler device for general medical use (laser Doppler MoorLab instrument VMS-LDF2, Moor Instruments Ltd., Axminster, UK) and a straight optic probe VP3 with a length of 10 mm, built to be used on the oral mucosa/teeth. The laser Doppler signal acquisition technique was performed according to our previous studies [27,28,29,30,31]. In order to stabilize the laser probe in the tooth’s cervical third, a double silicone impression was taken using Kit Optosil Comfort Putty and Xantopren Comfort Light, Haereus (Heraeus Kulzer, GmbH Leipziger Straße 2, Hanau, Germany). This is a silicone-based condensation curing material that takes impressions with high dimensional stability in moist dental environments, thus allowing precise reproduction detail (Figure 3). The impression was further used as an LDF probe holder for acquiring pulpal blood flow signals. After decontaminating the impression, a drill (drill bit 1.5 mm in diameter) was used to create a tunnel perpendicular to the cervical third of each tooth involved in the study, 1 mm in distance from the marginal gingiva, so as to avoid the bracket and the luting cement on the vestibular surface of the tested tooth. Thus, the laser Doppler probe was positioned only at the level of the unmodified enamel, avoiding the structurally modified enamel surface after removing the brackets and the adhesive material. To ensure the reproducibility of the laser Doppler signal acquisition, a guide mark was set on the fiber, which allowed it to be placed in the same position for each test. Additionally, during descimentation, as well as when removing the cement, the doctor performed these therapeutic maneuvers so as not to alter the surface of the enamel in the cervical area, where the laser Doppler probe was to be placed to acquire the signal from the pulp of the tooth.

Before making the LDF probe holder, the ceramic brackets were isolated with orthodontic wax (G.U.M. Ortho Wax, Sunstar, Etoy, Switzerland), which was applied to their retentive areas in order to protect them and allow the holder to be safetly removed from the oral cavity. Before LDF recordings, a heat-free light-cured liquid dam (LC Block-Out Resin, Ultradent, Products GmbH, Am Westhover Berg 30, Cologne, Germany) was applied around every tooth, using a of radius 3–4 mm (Figure 4). This liquid dam helped to reduce secondary signals generated by gingival microcirculation, as these may impair accuracy of the pulpal blood flow signal. Then, the tooth’s surface was cleansed and the LDF probe holder was positioned.

For each measurement, the pulp blood flow signal was recorded for 1 min and the evolution of vascular microdynamics at the dental pulp level was represented as pulsatory signals, as shown in Figure 5. The flow was related to the product of the average speed and concentration of mobile red blood cells in the tissue sample volume. The digital signal’s mean values and standard deviations were recorded for further analysis. A personal computer system (Dell, Intel Core i5, 7th gen, Round Rock, TX, USA) was used for data collection and processing data.

### 2.5. The Working Technique

After completion of the orthodontic treatment, the monocrystalline ceramic brackets fixed on the enamel of the vestibular surface of the frontal teeth were removed by one of the two therapeutic procedures: orthodontic forceps or laser-assisted, or using Er:YAG laser radiation with a wavelength of 2940 nm.

Each participant included in study signed an informed consent form. The patients’ levels of anxiety and fear of pain were evaluated before debonding, and the Wong–Baker FACES Pain Rating Scale was applied to evaluate their tooth pain perception immediately after debonding [21,22].

For each patient, six frontal maxillary teeth were considered, equally distributed on the left and right quadrants, as shown in Figure 4. For each participant, symmetric teeth were randomly assigned into one of the two debonding groups, using Er:YAG laser assisted action (Figure 6) or using orthodontic forceps (Figure 7).

For laser-assisted debonding, quadrant 1 was chosen (teeth: 1.3, 1.2, 1.1), and for conventional debonding (with orthodontic forceps), quadrant 2 was chosen (teeth: 2.1, 2.2, 2.3).

For the evaluation of the pulp vascular microdynamics, three paired assessments were made using LDF: at baseline (i.e., before the debonding procedure), immediately after debonding, and after 7 days. Additionally, the debonding time and the tooth sensitivity after debonding were recorded for both evaluated techniques. For this purpose, a digital timer and the Wong–Baker FACES Pain Rating Scale were used.

### 2.6. Data Analysis

For all numerical variables, the normality of the data distribution was checked with the Kolmogorov–Smirnov statistical test. For normally distributed data, parametric tests were further employed, such as the two-way ANOVA or paired *t*-test. For ordinal data (i.e., the ranks of the pain scores), non-parametric tests were applied, irrespective of their distribution.

The level of significance and confidence were 0.05 and 0.95, respectively. All reported probability values were two-tailed. For the sample size estimation, the threshold level of statistical power was 0.9.

Data analysis was conducted with the statistical software IBM SPSS v.20 and the R v.4.05 software packages (R Core Team, 2021; https://www.r-project.org/), including the packages “pwr2” v.1.0 and “MKpower” v.0.5.

#### 2.6.1. Analysis of the Wong–Baker Pain Rating Scale Pain Scores

Descriptive statistics included the mean (M) and standard deviation (SD). The Wilcoxon non-parametric statistical test was applied to compare the matched scores in the two groups.

#### 2.6.2. Analysis of the Working Time

Descriptive statistics included M and SD. The paired *t*-test was used to compare the matched times in the two groups. The variance of the data was compared with the Kolmogorov–Smirnov test.

#### 2.6.3. Analysis of the Laser Doppler Flowmetry Measurements

Descriptive statistics of the blood flux included M and SD for each meaningful combination of the categorical variables. Two-way ANOVA was applied to compare the blood flux mean values. At each of the three measurement times, based on the variance of the blood flow recording, the distribution of the flux variability across the treatments was compared with the Kolmogorov–Smirnov nonparametric test.

## 3. Results

The full raw data are presented in the Appendix A. Appendix A shows the responses to the pain questionnaire (pain scores) immediately after debonding and the time (s) necessary for the removal of monocrystalline ceramic brackets, using the two different techniques for debonding.

The descriptive statistics and the results of the Wilcoxon non-parametric statistical test for the data obtained using the Wong–Baker FACES Pain Rating Scale are presented in Table 1.

Thus, referring to the tooth sensitivity felt by the patient during debonding, the analysis of the recorded data (scores) showed a statistically significant difference between the two debonding methods, *p* < 0.001 (Table 1). The maximum tooth sensitivity score for the laser Er:YAG-assisted debonding was 2, while, for the conventional method, it was 6 (Appendix A).

The descriptive statistics and the results of the paired *t*-test for the recorded data of the time (s) required to remove the monocrystalline ceramic brackets are presented in Table 2.

The analysis of the data recorded for the time required to remove the brackets highlighted a statistically significant difference between the two debonding methods at *p* < 0.001 (Table 2).

Appendix A comprises the mean values of the pulpal blood flow of the 60 teeth included in the study, assessed by laser Doppler flowmetry in accordance with the study design.

Descriptive statistics of the blood flux, including M and SD for each meaningful combination of the categorical variables, are given in Table 3. The results of the two-way ANOVA applied to compare the blood flux mean values are also given.

These results did not show statistically significant differences (Table 3) between the two techniques.

## 4. Discussion

According to our study design, we recorded and evaluated three variables: tooth sensitivity associated with debonding, time required for the debonding, and pulp blood flow microdynamics after debonding, which was performed using Er:YAG laser radiation and compared to the conventional technique.

From the data analysis, the Er:YAG laser-assisted method produced significantly lower tooth sensitivity scores compared to conventional debonding. The subjective tooth sensitivity was reduced from a mean ± standard deviation of 3.07 ± 1.46 (on the Wong–Baker FACES Pain Rating Scale) when debonding was performed by the conventional method to 0.47 ± 0.86 when Er:YAG laser debonding was used. Based on these outcomes, the debonding procedure using laser Er:YAG radiation seems to be more comfortable for the patient for removing monocrystalline ceramic brackets.

For the time needed for debonding, the debonding method with Er:YAG laser radiation nedeed a significantly shorter working time than that required for the conventional method, which uses orthodontic forceps. The mean value of the time for laser-asissted debonding was 0.94 ± 0.254, and for the conventional debonding it was 1.64 ± 0.57. The necessary time for debonding was reduced by 0.697 ± 0.703 s per tooth when Er:YAG laser radiation was used. Therefore, according to the results of the present study, the debonding of monocrystalline ceramic brackets from the upper frontal human teeth is significantly faster when the Er:YAG laser is used than if the conventional method is used.

The results regarding the effect of the two debonding techniques on the pulp blood flow microdynamics showed that there was no difference in the blood microdynamics of the teeth after the debonding procedure (power of 0.9). After analyzing the recorded data for the three evaluated moments (i.e., before debonding, immediatelly after, and at 7 days after debonding—Table 3), statistically significant differences were noted only between the initial moment and after 7 days, for both debonding techniques (at *p* < 0.05). According to the data analysis, the blood flow mean value for the laser-asissted debonding group was 11.49 ± 2.66 PU before treatment and 16.27 ± 5.56 PU at 7 days after debonding. For the conventional debonding group, it was 12.26 ± 2.76 PU before treatment and 14.89 ± 4.39 PU at 7 days after debonding. Based on these results, there was no statistical difference between the two types of debonding methods in the pulpal blood flow changes as a result of the bracket removal manouvre.

Following the analysis of the specialized literature, we did not find other studies that considered the three variables evaluated in our current research. We found no data related to the three variables (i.e., tooth pain, debonding time, and pulp blood flow changes after debonding) included in our in vivo study, which were evaluated under the conditions of Er:YAG laser debonding compared to the conventional method.

Regarding removing brackets from the tooth enamel, in the specialized literature, other methods based on different types of chemical agents have been investigated. Khalil et colab. [12] used acetone and ethanol at different concentrations to test their ability to dissolve orthodontic adhesive; however, no statistically significant differences were found for shear strength and the residual adhesive index. Even when a longer application time was used, the results did not support their hypothesis. Because the Er:YAG laser has a thermal effect on tissues that contain water, the laser radiation can be absorbed directly by the adhesive resin containing the residual monomer, without creating an increased amount of heat [32].

However, the efficiency of laser equipment in the debonding process has been the subject of several studies, with several variables and techniques, for example, the type of lasers used, the same energy level or different levels, the type of brackets, adhesives, or the magnitude of the applied forces. Tozlu M. et al. [33] and Alakuş-Sabuncuoğlu et al. [34] evaluated the effect of laser radiation and proved that the Er:YAG laser (2940 nm) has the ability to be absorbed directly by the adhesive resin without negative consequences to the pulp tissues. Alakuş-Sabuncuoğlu et al. [34] studied the effects of the debonding of ceramic brackets with the Er:YAG laser by evaluating the shear strength and the residual adhesive index after the debonding procedure. The authors concluded that the use of the Er:YAG laser for the debonding of polycrystalline ceramic brackets is accompanied by a reduction in the debonding force and increased residual adhesive index scores.

Kim et al. [34] removed the monocrystalline brackets in less than 1 s by thermal ablation or photoablation and recorded a low temperature after debonding. They stated that the two mechanisms involved in debonding occur with small heat diffusion, which allows the tooth and the bracket to remain at temperatures close to physiological ones. These results were consistent with those of our study, in which a significant reduction in working time was observed when laser radiation was used, such that the mean value of time during laser-asissted debonding was 0.94 ± 0.254.

Referring to the type of bracket, ceramic brackets are made of alumina and are monocrystalline (sapphire) or polycrystalline structures. Monocrystalline ceramic brackets show higher strength, but the fracture strength is generally lower than that of polycrystalline high transmissivity brackets, thus increasing energy losses [35].

The procedure of removing the bracket from the surface of the tooth can cause a local irritation at the level of the dental pulp (thermal and/or mechanical in nature), which affects the microvascularization.

The removal of ceramic brackets using laser devices could represent a real alternative in debonding methods, especially in the case of the Er:YAG laser. Although the method presents numerous advantages, the effects of other types of lasers on the dental pulp have not yet been fully determined.

In our study, we opted for laser-assisted debonding with Er:YAG compared to the standard procedure currently used in most private practices, namely debonding with pliers that develop a mechanical force to “break” the adhesive at the bracket–tooth interface level.

With regards to the recording of vascular microdynamics at the level of the dental pulp, at the level of the marginal gingiva, and the oral mucosa, LDF is recognized as an objective, non-invasive evaluation method which provides continuous and real-time data [36,37,38,39,40].

Finding an optimal method for debonding ceramic brackets without destructive effects on enamel is of utmost importance. This would provide excellent insight for orthodontists, regarding the possibility of reducing the patient’s discomfort that usually accompanies this clinical maneuver.

Additionally, the significant reduction in working time can influence the sensitivity of the tooth subjectively perceived by the patient during debonding. Moreover, the significant difference between the pulp flow at seven days compared to before debonding may represent a possible evolution of vascular dynamics after the removal of the bracket from the surface of the tooth, irrespective of the force exerted on the tooth.

Therefore, based on these outcomes, obtained under the conditions of the current study, conducted in accordance with the established study design, the null hypothesis can be rejected. The debonding procedure using Er:YAG laser radiation produced significantly lower tooth sensitivity scores and was significantly faster than the conventional one for removing monocrystalline ceramic brackets. Regarding the vascular dynamics of the dental pulp, based on our results, the null hypothesis is accepted, which means we can say that there is no statistical difference between two types of debonding method related to the pulpal blood flow changes.

However, the following aspects could be considered as limitations of this study: the size of the sample, only considering upper frontal teeth in the study, and only using one wavelength for the laser debonding technique.

Moreover, other negative effects, mentioned in the specialized literature, during bracket removal were not investigated in the present study, including: the potential effects on the topography [10] and surface roughness of the enamel [41]; the effects on the removal of the adhesive layer (or adhesive remnant index) [42]; local irritation at the level of the dental pulp of thermal nature; [43] and mechanical alterations to the structure of the bracket [16].

## 5. Conclusions

Based on the results and taking into account the limits of this clinical study, we can conclude that laser Er:YAG-assisted debonding is a viable alternative to the conventional method for the removal of monocrystalline ceramic brackets.

Bracket debonding using laser Er:YAG radiation is faster and more comfortable for the patient than the conventional procedure, and there is no significant difference in pulp blood flow microdynamics between the two techniques.

However, future studies are needed for standardization of the working parameters of the laser equipment in order to apply the most effective laser-assisted debonding protocol in the daily practice of the orthodontists, which could represent real progress in the quality and efficiency of dental treatments.

## Figures and Tables

**Figure 1 ijerph-19-14564-f001:**
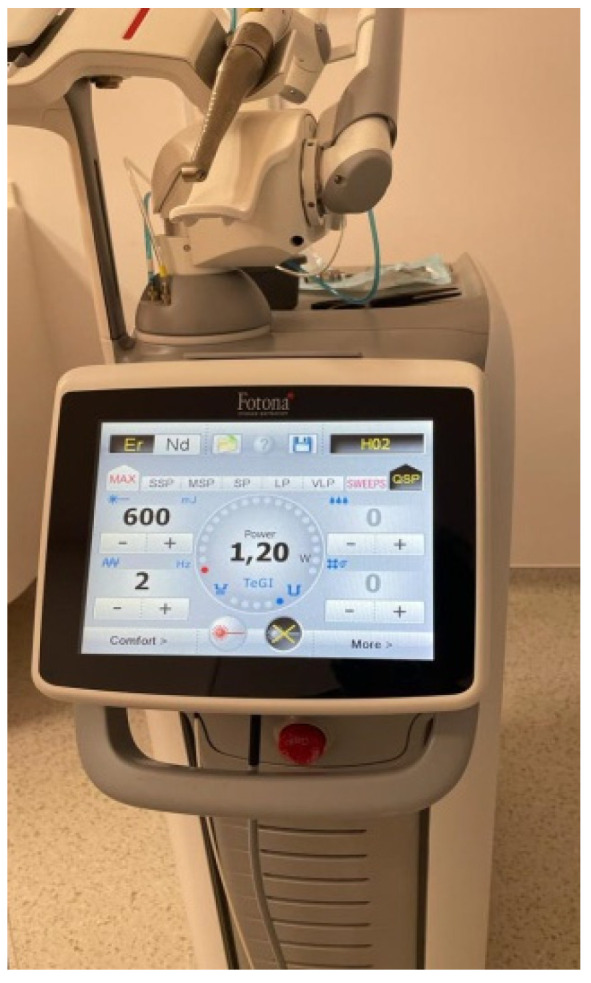
Er:YAG laser device (LIGHTWALKER AT S, M021-5AF/1 S, Fotona d.o.o., Ljubljana, Slovenia).

**Figure 2 ijerph-19-14564-f002:**
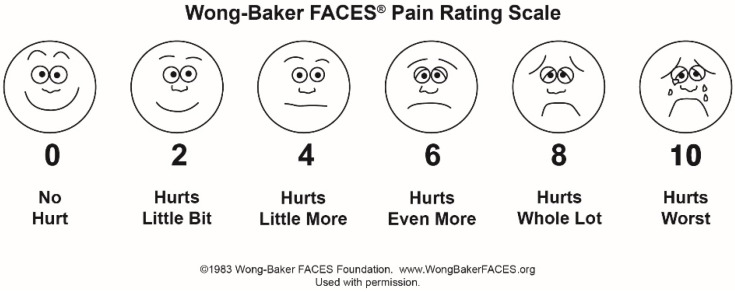
Wong–Baker FACES Pain Rating Scale based on facial appearance relative to pain levels (https://wongbakerfaces.org/wp-content/uploads/2016/05/FACES_English_Black.jpg, accessed on 11 October 2022). Used with permission.

**Figure 3 ijerph-19-14564-f003:**
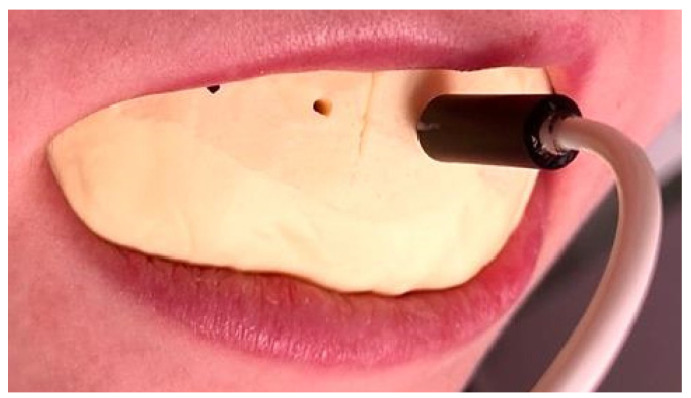
The LDF probe holder maintained the Doppler laser probe oriented towards the cervical third of the vestibular surface of the teeth selected in the study.

**Figure 4 ijerph-19-14564-f004:**
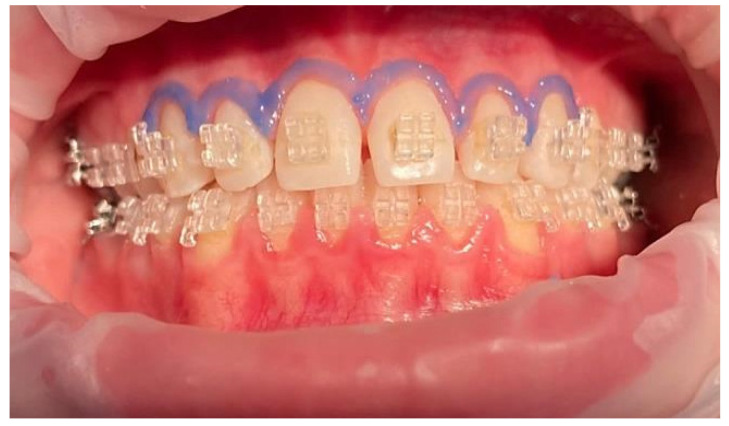
Clinical aspect of the dental units included in the study with the gingival barrier, before the debonding procedure.

**Figure 5 ijerph-19-14564-f005:**
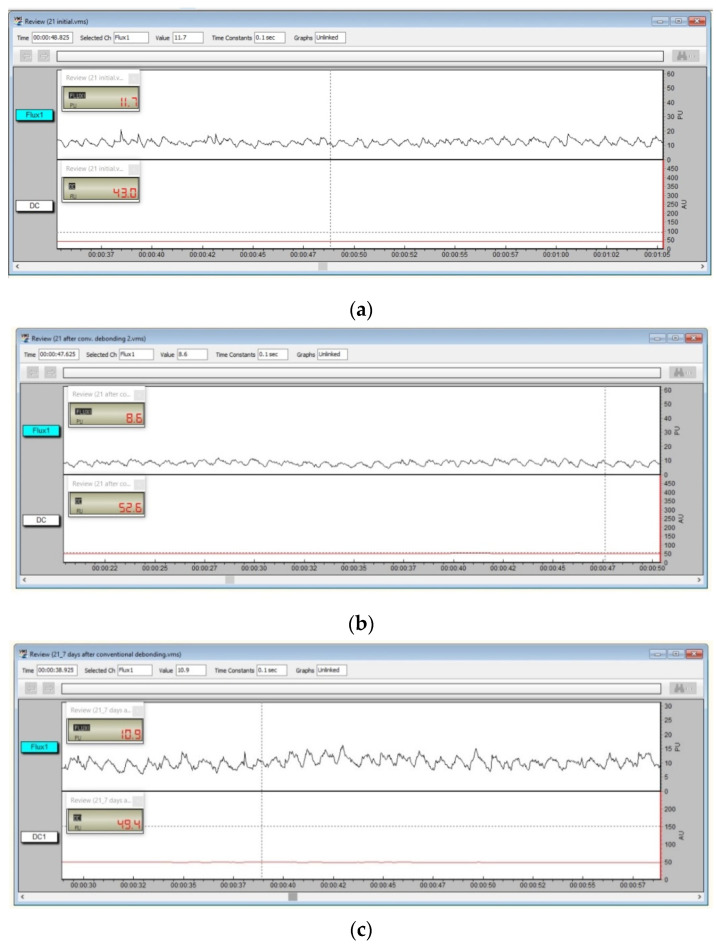
Laser Doppler pulp level signal for tooth 2.1. before debonding (**a**), immediately after conventional debonding (**b**), and seven days after the treatment (**c**).

**Figure 6 ijerph-19-14564-f006:**
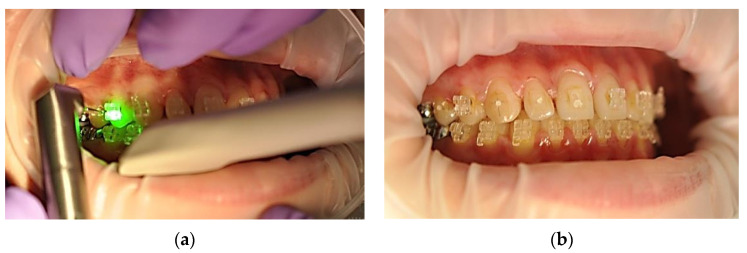
Application of laser Er:YAG radiation to dental units 1.3–1.1 (**a**) and clinical aspect after debonding (**b**).

**Figure 7 ijerph-19-14564-f007:**
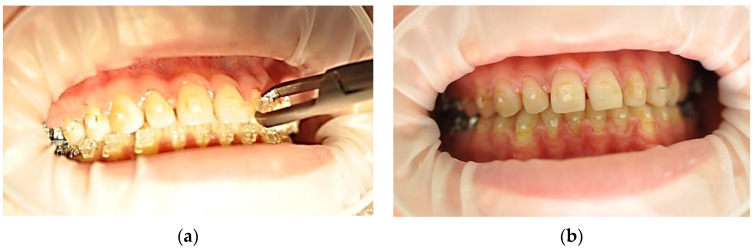
Use of the conventional method to dental units 2.1–2.3 (**a**) and clinical aspect after debonding (**b**).

**Table 1 ijerph-19-14564-t001:** The results of the statistical analysis of the Wong–Baker pain scores.

	Laser Er:YAG-Assisted ^a^	Conventional ^a^	*p*-Value ^a^
M ± SD	0.47 ± 0.86	3.07 ± 1.46	<0.001 **

^a^ Wilcoxon signed rank statistical test for the pain scores. Statistical significance, ** *p* < 0.001. Abbreviations: M, mean; SD, standard deviation.

**Table 2 ijerph-19-14564-t002:** The results of the statistical analysis of the working time.

	Laser Er:YAG-Assisted ^a^	Conventional ^a^	*p*-Value ^a^
M ± SD	0.94 ± 0.25	1.64 ± 0.57	<0.001 **
Difference between paired teeth: −0.697 ± 0.703; 95% CI (−0.959; −0.434)			

^a^ paired-*t*-test. Statistical significance, ** *p* < 0.001. Abbreviations: CI, confidence interval; M, mean; SD, standard deviation.

**Table 3 ijerph-19-14564-t003:** The results for the statistical analysis of the laser Doppler flowmetry measurements.

Time of Measurements	Laser Er:YAG-Assisted ^a^	Conventional ^a^	Two-Way ANOVA
Before debonding	11.49 ± 2.66	12.26 ± 2.76	Model: *p* < 0.160
			PatientID: *p* < 0.024 *
			Treatment: *p* = 0.243
After debonding	12.56 ± 3.94	12.10 ± 3.90	Model: *p* < 0.224
			PatientID: *p* < 0.289
			Treatment: *p* = 0.628
7 days after debonding	16.27 ± 5.56	14.89 ± 4.39	Model: *p* < 0.044 *
			PatientID: *p* < 0.017 *
			Treatment: *p* = 0.233

^a^ M ± SD. Statistical significance, * *p* < 0.05. Abbreviations: M, mean; SD, standard deviation.

## Data Availability

All raw data are provided as Appendix A.

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
