# Peer review of "Laser Er:YAG-Assisted Debonding May Be a Viable Alternative to the Conventional Method for Monocrystalline Ceramic Brackets"

_ijerph, 2022, doi:10.3390/ijerph192114564_

Round 1
Reviewer 1 Report
This study evaluated the effect of laser assisted debonding technique for removing the monocrystalline ceramic brackets from the human teeth, regarding the tooth sensitivity, working time and pulp blood flow microdynamics. Several studies have evaluated the use of Laser in the debonding of orthodontic brackets and promising results have been reported. The authors measured pulp blood flow, which was not available in other studies. However, other negative effects during bracket removal were not involved in the present study, including the potential effects on the topography and surface roughness of enamel, the effects on the removal of the adhesive layer (or adhesive remnant index), etc. These issues need to be addressed. The manuscript is generally well written, while it needs to be carefully polished to improve reading fluency. An unstructured “abstract” would be better. The discussion part needs to be improved regarding the logic and fluency.
Author Response
We would like to thank you for revising our article and for appreciating our work. We have taken into consideration your recommendations, therefore we have operated the following changes as can be seen in the attached answer.
Kind regards,
On Behalf of the Authors,
Mariana Miron

Reviewer 2 Report
The data are adequate in the paper, however, there are several flaws in the language, overall presentation and methodology. Please find my specific comments below:
1. It is suggested that the authors should get assistance for the language issue in order to bring a flow and coherence throughout the text.
2. It is suggested that the authors should merge all hypotheses in a single sentence.
3. Authors have stated the terms 'tooth sensitivity' and 'dental sensitivity' which cause a great confusion whilst reading the paper.
4. Why did the authors select 'Wong-Baker FACES' Pain Scale for the evaluation of tooth sensitivity? Authors should justify this selection scientifically as its applicability may be questioned for the evaluation of sensitivity of teeth.
5. The sections 2.3 and 2.4 should be placed as sections 2.1 and 2.2 so as to make a logical sequence in the text.
6. The discussion section includes unnecessary text for instance in lines: 302-304, it is stated that "The aim of this study was to evaluate laser Er:YAG assisted debonding technique compared to the conventional one, for removing the monocrystalline ceramic brackets from the human teeth". It is suggested that the authors should remove irrelevant text from the discussion in order to avoid repetition.
7. References and abstract should be provided in accordance with the journal's guidelines.
Author Response
Thank you for rigorously reviewing of our article and your detailed feedback which helps substantially in the improvement of our work. Consequently, we made the following changes which can be seen in the attached material.
Best wishes,
On Behalf of the Authors,
Mariana Miron

Round 2
Reviewer 1 Report
-
The author solved my doubts very well and the quality of the manuscript have been improved.
Author Response
Dear Reviewer 1,
We would like to thank you for your appreciation of our work, we have made the improvements also due to your suggestions and comments for which we are grateful.
Best regards,
On Behalf of the Authors,
Mariana Miron
Reviewer 2 Report
The authors have addressed most of my comments/suggestions and the paper looks better now.
In my previous review report, it was suggested that the authors should merge all hypotheses, however, in the current version of the paper, hypotheses have been completely removed. It would be much better if the authors would retain the hypothesis in the introduction section and either accept or reject it (as the case may be) in the discussion section.
Author Response
Dear Reviewer 2,
We would like to thank you again, for your careful reviewing of our manuscript. We have taken into consideration your suggestions, therefore we have operated the following changes as can be seen in the attached answer.
Kind regards,
On Behalf of the Authors,
Mariana Miron
